# Layerwise Recurrent Autoencoder for General Real-world Traffic Flow Forecasting

## Abstract

Accurate spatio-temporal traffic forecasting is a fundamental task that has wide applications in city management, transportation area and financial domain. There are many factors that make this significant task also challenging, like: (1) maze-like road network makes the spatial dependency complex; (2) the traffic-time relationships bring non-linear temporal complication; (3) with the larger road network, the difficulty of flow forecasting grows. The prevalent and state-of-the-art methods have mainly been discussed on datasets covering relatively small districts and short time span, e.g., the dataset that is collected within a city during months. To forecast the traffic flow across a wide area and overcome the mentioned challenges, we design and propose a promising forecasting model called *Layerwise Recurrent Autoencoder* (LRA), in which a three-layer stacked autoencoder (SAE) architecture is used to obtain temporal traffic correlations and a recurrent neural networks (RNNs) model for prediction. The convolutional neural networks (CNNs) model is also employed to extract spatial traffic information within the transport topology for more accurate prediction. To the best of our knowledge, there is no general and effective method for traffic flow prediction in large area which covers a group of cities. The experiment is completed on such large scale real-world traffic datasets to show superiority. And a smaller dataset is exploited to prove universality of the proposed model. And evaluations show that our model outperforms the state-of-the-art baselines by 6% - 15%.

## 1 Introduction

Spatiotemporal traffic flow forecasting task is currently under a heated discussion and has attracted a large research population. The application of this task is wide, including transportation anomaly detection, optimal resource allocation, logistic supply chain and city management. However, since the dynamic environment of traffic condition and the inherent complexity of large scale forecasting tasks, the task is challenging (Drew, 1968). In this paper, we investigate the advantages from current methods and propose a model that can solve the task with spatiotemporal modeling. Even in the dataset with large road network, the model works well. The goal of the traffic flow forecasting is to predict the future traffic flow in the whole road network with the input sequences from sensors and the space correlations of those sensors.

The main obstacle of traffic flow prediction task is to find the appropriate spatiotemporal dependencies (Atwood & Towsley, 2016). For two reasons. First, the time series of traffic flow is dynamic, where the rush hours in the morning and evening cause a non-linear variate on the flow, and the information in different days of the week incurs more complex relationships. Second, the space correlations between sensors in the road network are difficult to be determined. Figure 1 demonstrates an example of the complexity in spatial dependency modeling. Point A and Point B are two sensors in a freeway network, and their geographic distance is close, but the driving distance is much farther than it seems to be. Besides, since they are deployed on the opposite sides of the road, the flows are different a lot. This instance illustrates that the spatial distance is not supposed to be Euclidean, but to be dominated by the road topology.

To overcome the challenges, we propose a deep-learning based *layerwise recurrent autoencoder* (LRA) for sequence-to-sequence traffic flow forecasting. The contributions of this paper are summarized as the following:

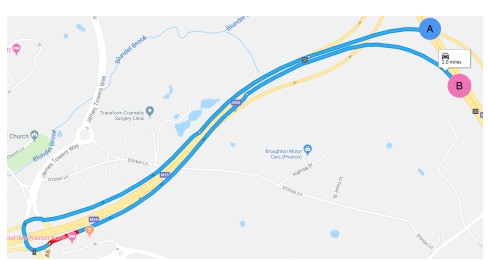 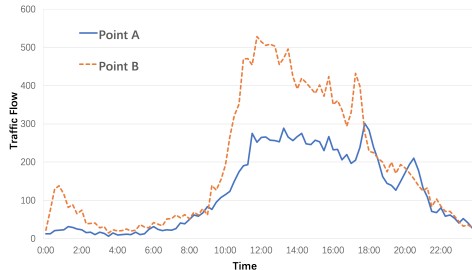

Figure 1: Distance dependency is not directional and geographic. The Point A and B are geographically close, but they stand on opposite sides of the road, the driving distance is far. And their traffic flows are different a lot, shown in the right figure.

- Originally use a very large scale traffic dataset[1], which covers three cities, and a comparatively small dataset to evaluate our model with common used baselines. The results of experiments prove the generalization and effectiveness of LRA.

- Creatively employ three-layer time series sequences (flow sequence of current time, flow sequence of this time of last week and last four weeks) as inputs of LRA, from which LRA obtains the knowledge of time series relationships and the periodical effect of traffic flows.

- Innovatively exploit the driving distances between sensors in the road network to model spatial dependency, which is presented as a directed graph whose nodes are sensors and edge weights are spatial correlations.

## 2 RELATED WORK

The history of traffic forecasting has been decades long, and many methods have emerged. Especially in recent years, the instruments and infrastructures of sensors are developed, these detectors provide the possibility of accurate record of traffic volume within transportation network. The methods on this subject can be mainly divided into two categories: classic statistical approaches and data-driven approaches. The classical time-series approaches are mainly based on queuing theory and statistic theory (Cascetta, 2013). While the data-driven methods focus on curriculum learning and have recently attracted plenty attentions.

In this paper, some cutting-edge models and popular accepted methods are studied from both categories, but these methods are found often share similar problems in different experimental settings and face some limitations when apply with real-world complex tasks. In the category of classic methods, Box & Jenkins (1970) acts as a fundamental role in this area of forecasting by generating the model called autoregressive moving average (ARMA) model. Taking ARMA as basis, an integrated version of ARMA for traffic forecasting is built, we cite as autoregressive integrated moving average (ARIMA) (Box et al., 2015; Moorthy & Ratcliffe, 1988; Lee & Fambro, 1999). The ARIMA model is a general extension of ARMA, and starting from ARIMA, a bunch of variations born, including seasonal ARIMA (SARIMA) (Williams & Hoel, 2003), which is designed for capturing the common periodical features from many time-series processes and space-time ARIMA (STARIMA) (Williams & Hoel, 2005), which models for short-term traffic flow forecasting in multiple nodes within a transportation topology. These statistical methods are primarily based on queuing theories and mathematical simulations, it is hard for them to learn patterns from dynamics and complexities. As a result, though they perform satisfactorily on short-term and small-scale datasets in some research areas (Lippi et al., 2013), when apply to real-time large-scale scenarios, their performance is barely satisfactory (Cheng et al., 2017). In this paper, we select the two most representative methods, ARMA and ARIMA as the baselines.

In data-driven learning community, over the recent decade, a number of models are built by neural networks, and have gotten high performances that surpass the traditional time analysis methods. The works in Laptev et al. (2017); Yu et al. (2017), apply recurrent neural networks (RNNs) to

---

[1]The dataset is from http://tris.highwaysengland.co.uk/detail/trafficflowdata

study time series prediction. And convolutional neural networks (CNNs) are also chosen for flow forecasting in Ma et al. (2017); Zhang et al. (2017). Besides, the echo state networks (ESNs) (Ilies et al., 2007) are deployed on some light applications for forecasting tasks. However, because of the mentioned complexities and challenges in traffic forecasting tasks, the unsolved problem is to design a general approach with temporal and spatial correlations modeling. In Lv et al. (2015), the authors use stacked autoencoder (SAE) model to learn generic time series features, and the model is applied using autoencoder as building block to represent traffic flow features for prediction. See in Yu et al. (2016), a temporal regularized matrix factorization method is proposed and find graph regularization connections to learn the dependencies, but this model pays insufficient attention on nonlinear temporal relationships. Other researchers exploit latent space models for traffic volume prediction, while the distance dependency is extracted by controversial geographical distances (Deng et al., 2016; Sun et al., 2006) or by a rough epitome of region flows (Zhang et al., 2016). In Li et al. (2018), the authors obtain the space-time dependencies with diffusion convolutional recurrent neural networks (DCRNN), the distance correlations are represented as a directed graph and the model relates traffic flow to a diffusion process. In this work, we select ESNs, SAE, RNNs and DCRNN as the representatives of data-driven models, the comparison between these models and the proposed one is shown in the latter part of this paper, the generalization and effectiveness of our model is also demonstrated.

## 3 LAYERWISE RECURRENT AUTOENCODER

In this section, the structure of LRA is introduced by order, and the structure for spatial-temporal modeling is formulated.

### 3.1 TEMPORAL DEPENDENCY MODELING

To extract temporal relationships within the history traffic flows, we model this process as a layering structure with autoencoder as cell. An autoencoder is used to reproduce its inputs, in other words, the target output of autoencoder is its input. The structure of autoencoder is shown in Appendix B. With sequences of traffic flows $\{\boldsymbol{x}^{(1)}, \boldsymbol{x}^{(2)}, \boldsymbol{x}^{(3)}, ...\}$ as input, an autoencoder first encodes the input $\boldsymbol{x}^{(i)}$ to a hidden representation, and then decodes the representation back to a reconstruction. To minimizing reconstruction error $L(\boldsymbol{X}, \boldsymbol{Z})$, where $\boldsymbol{X}$ is the input matrix of the autoencoder and $\boldsymbol{Z}$ is the output matrix, we denote it as $\theta$, as

$$\theta = arg\min_{\theta} L(\boldsymbol{X}, \boldsymbol{Z}) = arg\min_{\theta} \frac{1}{2} \sum_{i=1}^{N} \left\| \boldsymbol{x}^{(i)} - \boldsymbol{z}\left(\boldsymbol{x}^{(i)}\right) \right\|^{2}, \tag{1}$$

where N is the length of the input sequence, and $\boldsymbol{z}(\cdot)$ is the reconstruction.

When take the sparsity constrains into consideration, to achieve the sparse representation in hidden layer (Lv et al., 2015), we minimize the reconstruction error as

$$\text{SAO} = L(\boldsymbol{X}, \boldsymbol{Z}) + \gamma \sum_{j=1}^{H_D} \text{KL}\left(\rho \| \hat{\rho}_j\right), \tag{2}$$

where $\gamma$ is the weight of the sparsity term, $H_D$ is the number of hidden units, $\rho$ is a sparsity parameter and is typically a small positive value, $\hat{\rho}_j$ is the average activation of hidden units, and $\text{KL}(\rho \| \hat{\rho}_j)$ is the Kullback-Leibler (KL) divergence, which provides the sparsity constraints on the coding, defined as

$$\text{KL}(\rho \| \hat{\rho}_j) = \rho \log \frac{\rho}{\hat{\rho}_j} + (1 - \rho) \log \frac{1 - \rho}{1 - \hat{\rho}_j}. \tag{3}$$

The SAE is created by hierarchically stacked autoencoders, in which the input of the $k$th layer is the output of the $(k-1)$th layer and a logistic regression is on the top. The structure of SAE is illustrated in Appendix B. In this paper, to extract more detailed temporal relationships in traffic history, we employ a three-layer SAE, with three-layer input as the flow sequences of current time, flow sequences of this time of last week and last four weeks relatively.

Following the layering SAE model, we employ a sequence-to-sequence RNNs structure. At training time, we feed ground truth into this RNNs architecture. And during testing stage, the ground truth values are replaced by the output of SAE model for forecasting. The backpropagation algorithm is used to optimize this process.

To avoid the vanishing gradient problem in long lasting dataset with the traditional RNNs models, we use long short term memory (LSTM) (Gers et al., 2002) in our model. The key of LSTM model is memory cell, which allows LSTM to remove or maintain the information, with special structures called gates in every memory cell, including input gate, forget gate and output gate. The memory cells can help for remembering the temporal relationships from SAE model and outperform other RNNs models when competing on large-scale long-span datasets (Seo et al., 2016).

## 3.2 Spatial Dependency Modeling

The correlations of spatial dependency are complex, and even more abstract than the temporal relationships especially in large-scale road networks, in this paper, we model the spatial dependency between sensors as a directed graph and analysis the relationships with CNNs.

As for the directed graph, in which takes sensors as nodes and driving distance as edges. We denote the graph as $\mathcal{G} = \langle \mathcal{V}, \mathcal{E} \rangle$. $\mathcal{V}$ is the set of nodes and $\mathcal{E} \subseteq \{(u, v) | u \in \mathcal{V}, v \in \mathcal{V}\}$ is the set of edges. In the graph $\mathcal{G}$, the distribution of the sensors is viewed as a space matrix, with inner elements are weights between each other. The space matrix is denoted as

$$\boldsymbol{M} = \begin{bmatrix} m_{11} & m_{12} & \cdots & m_{1n} \\ m_{21} & m_{22} & \cdots & m_{2n} \\ \vdots & \vdots & \ddots & \vdots \\ m_{n1} & m_{n2} & \cdots & m_{nn} \end{bmatrix}$$

where $m_{ij}$ is the weight from $v_i$ to $v_j$.

With the structured space matrix, the right way to express and exploit the dependency leads improvements in prediction performance. We encode the space matrix $\boldsymbol{M}$ with a graph convolution networks (GCNs) model (Kipf & Welling, 2017) to extract spatial dependency for helping traffic predict (Atwood & Towsley, 2016). The core work of the GCNs is to map from the input $\boldsymbol{M}$ to the convolutional representation that records the influential index of each sensors. After training epochs, the output matrix converges to a concise distribution $\boldsymbol{P}$, the $i$th row in $\boldsymbol{P}$ is the space correlation vector of node $v_i \in \mathcal{V}$.

The convolutional layerwise propagation in our paper is defined as

$$\boldsymbol{H}^{(l+1)} = \sigma\Big( \tilde{\boldsymbol{D}}^{-\frac{1}{2}} \boldsymbol{M} \tilde{\boldsymbol{D}}^{-\frac{1}{2}} \boldsymbol{H}^{(l)} \boldsymbol{W}^{(l)} \Big), \tag{4}$$

where $\tilde{\boldsymbol{D}}_{ii} = \sum_j \boldsymbol{M}_{ij}$ and $\boldsymbol{W}^{(l)}$ is a trainable matrix. While $\sigma(\cdot)$ is activation function, we use ReLU$(\cdot)$ in this work. $\boldsymbol{H}^{(l)}$ denotes the matrix of last layer and $\boldsymbol{H}^{(0)}$ is input matrix $\boldsymbol{M}$.

We also consider convolutions with a filter $g_\theta = \text{diag}(\theta)$ ($\theta \in \mathbb{R}^N$ in the Fourier Domain) in GCNs as the multiplication of element $x \in \mathbb{R}^N$ (Defferrard et al., 2016), defined as

$$g_\theta \star x = \boldsymbol{U} g_\theta \boldsymbol{U}^\top x, \tag{5}$$

where $\boldsymbol{U}$ is the matrix of eigenvectors of the normalized graph Laplacian $\boldsymbol{L} = \boldsymbol{I}_N - \tilde{\boldsymbol{D}}^{-\frac{1}{2}} \boldsymbol{M} \tilde{\boldsymbol{D}}^{-\frac{1}{2}} = \boldsymbol{U} \boldsymbol{\Lambda} \boldsymbol{U}^\top$ with a diagonal matrix of eigenvalues $\boldsymbol{\Lambda}$, $\boldsymbol{U}^\top x$ is the Fourier transform of the element $x$. Analysis on Equation 5 indicates that computational cost on computing the eigenvalue decomposition of $\boldsymbol{L}$ is expensive. To promote efficiency, we exploit a truncated presentation as

$$g_{\bar{\theta}}(\boldsymbol{\Lambda}) \approx \sum_{k=0}^{K} \bar{\theta}_k T_k(\hat{\boldsymbol{\Lambda}}), \tag{6}$$

where $\hat{\mathbf{\Lambda}} = \frac{2}{\lambda_{max}}\mathbf{\Lambda} - \boldsymbol{I}_N$, $\lambda_{max}$ means the largest eigenvalue of $\boldsymbol{L}$. While $T_k(\cdot)$ is recursively defined as $T_k(x) = 2xT_{k-1}(x) - T_{k-2}(x)$, with $T_0 = 1$ and $T_1 = x$. See Hammond et al. (2011) for more discussion of the truncation.

Combine the knowledge of Equation 5 and 6, the filter is substituted by $g_{\bar{\theta}}$, and we have

$$g_{\bar{\theta}} \star x \approx \sum_{k=0}^{K} \bar{\theta}_k T_k(\hat{\boldsymbol{L}})x, \tag{7}$$

where $\hat{\boldsymbol{L}} = \frac{2}{\lambda_{max}}\boldsymbol{L} - \boldsymbol{I}_N$. And the complexity of the computation has decreased from $\mathcal{O}(N^2)$ in Equation 5 to $\mathcal{O}(N)$ in Equation 7.

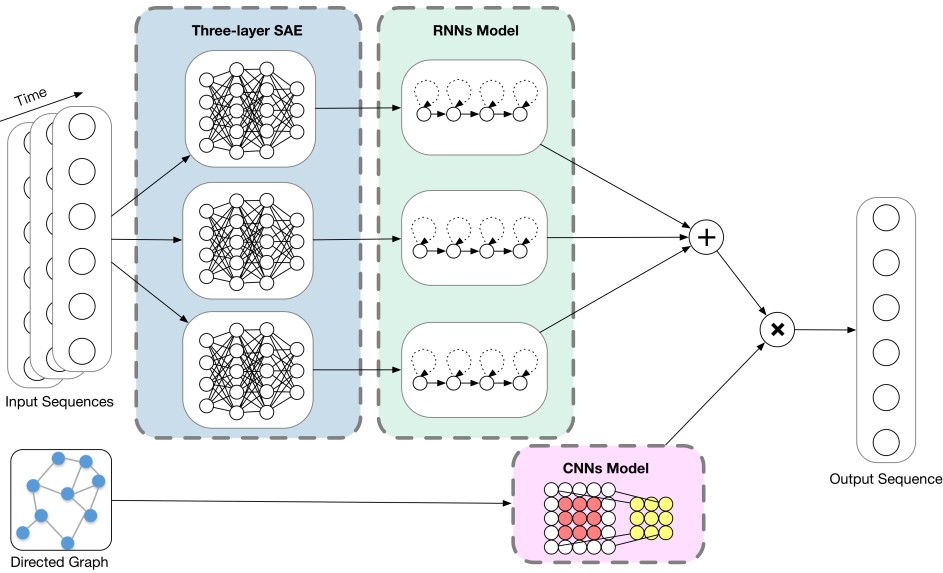

Figure 2: The architecture of LRA, designed for general traffic flow forecasting. The temporal relationships are extracted by three SAEs, whose outputs are fed to RNNs model for prediction. Then the final states of RNNs model and CNNs model are merged, and system output sequence is generated. All models are trained by minimizing the cross-entropy loss by backpropagation through time.

With spatiotemporal modeling and RNNs model, LRA is built with three parts for extracting spatial-temporal dependencies, whose architecture is shown in Figure 2. The whole network is trained to minimize the loss value of generated output sequence by backpropagation algorithm.

## 4 EXPERIMENT

In this paper, we complete the experiments with two real-world datasets: (1) **ENG-HW**: This dataset contains traffic flow information from inter-city highways between three cities, recorded by British Government. We conduct the experiments with 249 sensors and collect a whole year of data ranging from January 1st 2014 to December 31st 2014. (2) **ST-WB**: This traffic dataset is collected by SenseTime and Shanghai West Bund Development (Group) Co., Ltd. We select 220 sensors in the Shanghai West Bund area and collect 2 months of data for the experiments, ranging from July 1st 2018 to August 31st 2018. Compared with the first one, this dataset is relatively small-scale and simple, which is used for test universality of our model on both large and small datasets. On both datasets, we slice traffic flow information into 15 minutes windows, where 70% of data is for training, 10% for validation and remaining 20% for testing. The distribution of the sensors in ENG-HW dataset and more details of the two datasets are illustrated in Appendix C.

Table 1: Performance comparison between LRA and other approaches for traffic flow forecasting. From the results of our experiments, LRA has the best performance with all metrics on both large and small scale datasets.

| | $T$ | Metric | ARMA | ARIMA | ESNs | SAE | LSTM | DCRNN | **LRA** |
|---|---|---|---|---|---|---|---|---|---|
| ENG-HW | 15 min | MAE | 40.88 | 40.17 | 33.43 | 30.22 | 25.67 | 24.82 | **22.21** |
| | | RMSE | 69.81 | 68.23 | 48.25 | 52.52 | 43.93 | 42.26 | **40.11** |
| | | MAPE | 15.9% | 15.1% | 15.8% | 15.7% | 15.2% | 14.3% | **12.5%** |
| | 30 min | MAE | 41.44 | 42.98 | 35.08 | 35.12 | 28.34 | 29.27 | **25.19** |
| | | RMSE | 74.49 | 73.24 | 56.06 | 59.66 | 50.91 | 50.19 | **43.97** |
| | | MAPE | 18.4% | 18.6% | 17.2% | 16.3% | 16.7% | 16.6% | **14.4%** |
| | 60 min | MAE | 54.89 | 58.45 | 50.19 | 42.34 | 37.78 | 32.47 | **27.58** |
| | | RMSE | 78.65 | 78.62 | 59.76 | 61.47 | 53.21 | 55.76 | **49.09** |
| | | MAPE | 22.3% | 20.1% | 20.2% | 17.9% | 17.4% | 17.7% | **15.9%** |
| ST-WB | 15 min | MAE | 48.32 | 50.66 | 42.11 | 41.20 | 34.99 | 30.56 | **3.14** |
| | | RMSE | 82.23 | 80.88 | 75.90 | 70.97 | 68.23 | 65.44 | **55.46** |
| | | MAPE | 9.23% | 8.97% | 8.02% | 7.87% | 7.23% | 7.10% | **6.04%** |
| | 30 min | MAE | 54.68 | 58.34 | 48.50 | 46.88 | 38.09 | 36.62 | **34.39** |
| | | RMSE | 91.72 | 92.02 | 80.69 | 78.33 | 73.76 | 69.88 | **63.34** |
| | | MAPE | 10.6% | 9.83% | 9.52% | 8.20% | 7.46% | 7.58% | **6.93%** |
| | 60 min | MAE | 90.25 | 89.45 | 50.33 | 47.96 | 42.54 | 41.22 | **37.75** |
| | | RMSE | 99.14 | 102.4 | 92.55 | 94.43 | 89.73 | 74.62 | **68.73** |
| | | MAPE | 12.5% | 10.9% | 11.3% | 9.43% | 8.99% | 8.67% | **8.05%** |

As for another input of our model, road topology information, we compute the directed road network of sensors, where the distances are different between two sensors in different directions, and a space matrix is generated with threshold Gaussian kernel, as

$$m_{ij} = \left( - \frac{\text{dist}(v_i, v_j)^2}{\sigma^2} \right), \tag{8}$$

where $m_{ij}$ is the element in space matrix $\boldsymbol{M}$, represents the edge weight from $v_i$ pointing to $v_j$, $\text{dist}(v_i, v_j)$ is the directed driving distance between $sensor^i$ and $sensor^j$, $\sigma$ denotes the standard deviation of distances and if $\text{dist}(v_i, v_j)$ is less than threshold $\kappa$, we regard $m_{ij} = 0$.

## 4.1 EXPERIMENTAL SETTINGS

In the experiments, we compare the performance of LRA with popularly used methods and state-of-the-art model, including: (1) **ARMA**: which provides a parsimonious description of a weakly stationary stochastic process, consists of two polynomials, one for autoregression and the second for moving average. (2) **ARIMA**: which is widely used in statistics and econometrics, especially in time series analysis. (3) **ESNs**: a kind of RNNs model with a sparsely connected hidden layer, which is fixed and randomly assigned. (4) **SAE**: a deep neural network model that uses autoencoder as cell, good for time series forecasting tasks since the capability of extracting temporal dependency. (5) **LSTM**: a variant of RNNs model which is popular for classifying, processing and predicting tasks based on time series data. (6) **DCRNN**: one of the cutting edge deep learning models for forecasting, which uses a diffusion process during training stage to learn the representations of spatial dependency.

We built all the above neural network based models by Tensorflow (Abadi et al., 2016). We record the detail settings in Appendix D.

## 4.2 EXPERIMENT RESULTS OF TRAFFIC FLOW FORECASTING

The algorithms are evaluated by three popularly accepted metrics in transportation area, including (1) Mean absolute error (MAE), is known as a scale-dependent accuracy metric and is common in time series analysis. (2) Root mean squared error (RMSE), is frequently used for measuring the differences between prediction value and ground truth. (3) Mean absolute percentage error (MAPE),

is a measure of prediction accuracy in forecasting areas, usually expresses as a percentage. Note that comparisons across different datasets are invalid, since all the three metrics are scale-dependent. Formulations of the three measurements, see Appendix E.

Table 1 records the performances of different methods for three forecasting horizons in two datasets. From the table, we notice the following facts that: (1) The deep learning based methods, including ESNs, SAE, LSTM, DCRNN and LRA, outperform the statistical methods. (2) The deeper and more complex models are supposed to perform better than lighter ones, however there is an exception that the performances of DCRNN and LSTM are compatible, we guess it may because that the diffusion convolutional layers in DCRNN extracts insufficient spatial correlations in the experiments. (3) LRA achieves the best performance regardless metrics and datasets, which reflects the generalization and effectiveness of the proposed model. Besides, the forecasting task on the ENG-HW dataset is more difficult than that on ST-WB dataset, since the scale of ENG-HW dataset is larger and spatiotemporal dependencies are more complex. As the consequence, we use ENG-HW as the default dataset in the following discussion.

### 4.3 EXPERIMENT RESULTS OF SPATIAL AND TEMPORAL DEPENDENCY

In this part, we design experiments to proof the effect of spatial and temporal dependencies modeling by comparing the performance between LRA with two variants: (1) LRA-NoSAE, which feeds input time series sequences directly to RNNs model (the green part in Figure 2) in the LRA and cancels encode-decode process of SAE model, this variant system gets fewer time relationships from inputs. (2) LRA-NoConv, which abandons CNNs model (the pink part in Figure 2) in the LRA, the outputs of RNNs model become the final output sequences, this mutation of LRA gets less sensitive to space correlations. Figure 3 shows the learning curve of the above two variants and LRA with regard of MAE, we keep the parameters of all three models as similar as possible. From the learning curve, LRA reaches the lowest MAE value, meanwhile, LRA-NoConv has a much higher MAE value, which illustrates the effect of our spatial dependency modeling. Besides, the learning curve of LRA-NoSAE almost gets the same level of the LRA, but the speed of convergence is much slower, this fact proves the effect of time relationship modeling. The Table 2 shows the comparison results of these three models and their convergent speed. Combine the observations of Figure 3 and Table 2, the superiority of spatial dependency modeling is proved for helping promote accuracy of prediction, while the aim of temporal dependency modeling is to boost the training progress.

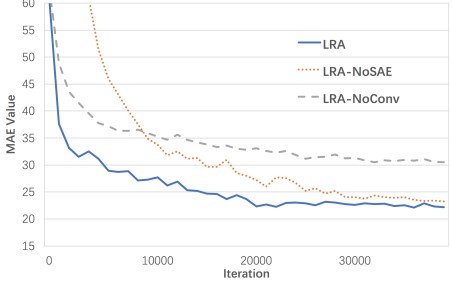

Table 2: Comparison for LRA, LRA-NoSAE and LRA-NoConv on the dataset ENG-HW in prediction horizon of 15 min. Note that the *Convergence Speed* means the number of iteration before models get convergent state (numerical fluctuation $< 10\%$).

|  | MAE | Convergence Speed |
|---|---|---|
| LRA | **22.21** | **17000** |
| LRA-NoSAE | 23.27 | 26000 |
| LRA-NoConv | 30.49 | 17000 |

Figure 3: Learning curve of LRA, LRA-NoConv and LRA-NoSAE on the ENG-HW dataset.

### 5 CONCLUSION

Traffic flow forecasting is an essential problem in many areas. There have been some methods performs well in specific conditions, however, a universal method for such problem, especially in large-scale road network, is absent. In this paper, we modeled the spatial-temporal dependencies and formulated such task by proposing the *layerwise recurrent autoencoder* (LRA) model. This model originally uses driving distance for modeling space dependencies and works well for general flow prediction. Meanwhile, the superiority and universality of our model are evaluated on both large and small real-world datasets with comparison to other common and state-of-the-art baselines. For the future work, we will investigate the following topics: (1) adding weather factors into LRA model for

more accurate prediction; (2) implementing the proposed model to other spatiotemporal forecasting tasks, e.g., pedestrian volume forecasting and audience distribution prediction.

ACKNOWLEDGMENTS

This project is fully funded by SenseTime Group Limited, and the ST-WB dataset is cooperatively collected by SenseTime and Shanghai West Bund Development (Group) Co., Ltd.

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

## APPENDIX

## A NOTATION

Table 3: The main notations used in the paper

| *Name* | |
|---|---|
| $\mathcal{G}$ | a directed graph |
| $\mathcal{V}, v_i$ | the set of sensors in the graph and the $i$th sensor |
| $\mathcal{E}$ | the set of weights in the graph |
| $\boldsymbol{M}$ | a space matrix |
| $\boldsymbol{H}^{(l)}$ | the $l$th layer of GCNs |
| $\boldsymbol{W}^{(l)}$ | the trainable matrix of $l$th layer in GCNs |
| $\boldsymbol{L}$ | normalized graph Laplacian |
| $\boldsymbol{U}, \boldsymbol{\Lambda}$ | the eigenvector matrix and eigenvalue matrix of $\boldsymbol{L}$ |

## B  THE STRUCTURE OF AUTOENCODER AND SAE

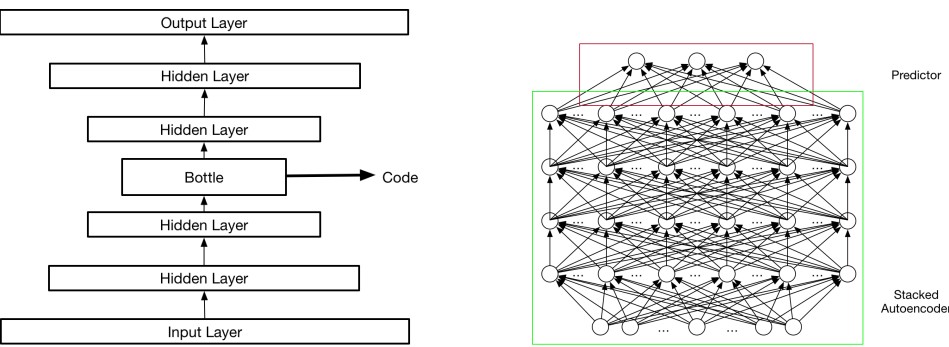

Figure 4: The structure of autoencoder (left) and SAE (right)

## C  DATASET

We use two real-world datasets for experiments:

- **ENG-HW** This traffic dataset is collected by British Government, which covers inter-city freeways between three cities, including Manchester, Liverpool and Blackburn. We select 249 sensors and collect one year of data ranging from January 1st 2014 to December 31st 2014 for the experiments. The total number of the piece of data is 8,724,960, the mean value of this dataset is 466.

- **ST-WB** This traffic dataset is collected by SenseTime and Shanghai West Bund Development (Group) Co., Ltd. We collect 220 sensors and collect two months of data ranging from July 1st 2018 to August 31st 2018 for the experiments. The total number of the piece of data is 3,928,320, the mean value of this dataset is 972.

In the both datasets, we slice time window to 15 minutes, and 70% data is used for training, 10% for validation and remaining 20% for testing. The distribution of sensors in ENG-HW dataset is shown in Figure 5.

## D  THE DETAILED EXPERIMENTAL SETTING

**ARMA**    Autoregressive moving average model, where the lag of AR is set to 3, the lag of MA is 0. The model is implemented by statsmodels python package.

**ARIMA**    Autoregressive integrated moving average model, in which the number of the AR lag is 3, the number of integrated term is set to 2, and MA lag is 0. The model is implemented by statsmodels python package.

**ESNs**    Echo state networks with a reservoir pool that holds 500 neurons, and the sparsity of the pool is assumed to be 5%; the leaking rate is set to 0.2, spectral radius is equals 0.9.

**SAE**    Stacked autoencoder with three hidden layers, each contains 800 cells, the learning rates are set to $1e^{-3}, \frac{1}{3}e^{-3}$ and $1e^{-4}$ for three hidden layers. The model is trained with batch size 128.

**LSTM**    Long short term memory with two hidden layers, within each holds 200 memory cells, L1 weight decay is $2e^{-4}$ and L2 weight decay is $2e^{-5}$. The model is trained with batch size 128.

**DCRNN**    Diffusion convolutional recurrent neural networks, the setting is the same as its authors recommend in `https://github.com/liyaguang/DCRNN`.

**LRA**    Layerwise recurrent autoencoder. In each SAE model, there are three hidden layers with 800 units, the learning rates are set to $1e^{-3}, \frac{1}{3}e^{-3}$ and $1e^{-4}$ for three hidden layers. In the RNNs model, there are two hidden layers, each layer has 200 LSTM cells, with L1 and L2 weight decay

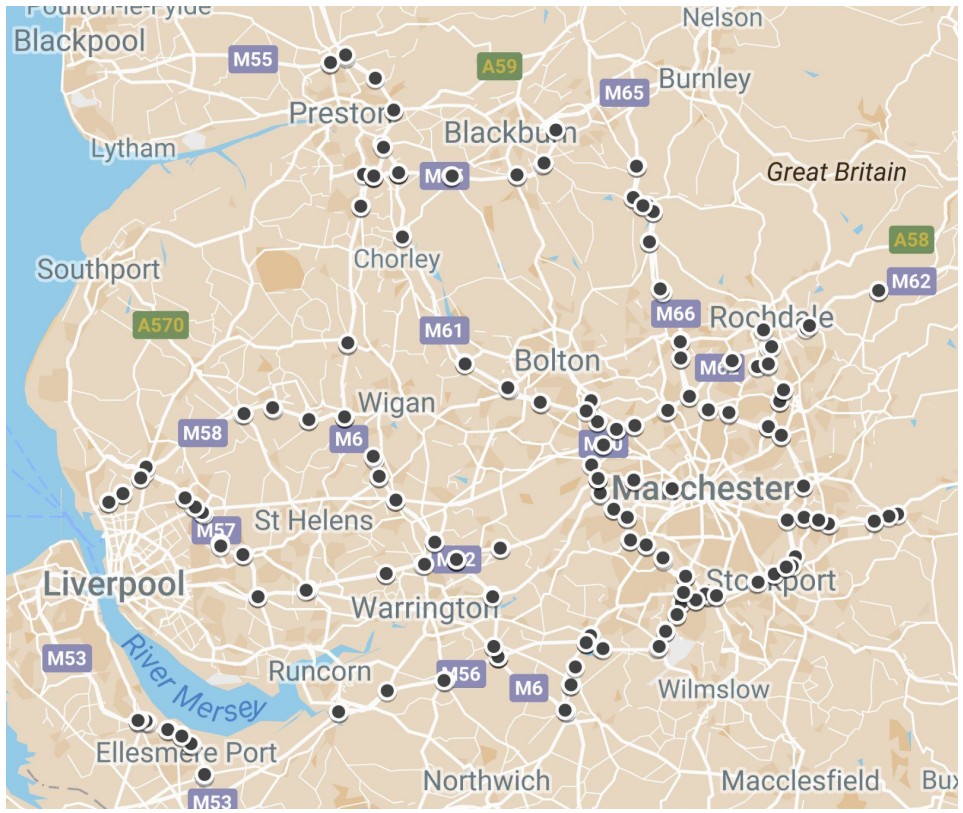

Figure 5: The distribution of sensors in the ENG-HW dataset.

are $2e^{-4}$ and $2e^{-5}$. As for CNNs model, the number of the node regards to the number of sensors in the experiments, the number of feature in each node is 1 and number of class is ignorable.

## E    METRICS

Denote $\boldsymbol{x} = \left(x^{(1)}, x^{(2)}, \ldots, x^{(n)}\right)$ is the ground truth of traffic flow, and $\hat{\boldsymbol{x}} = \left(\hat{x}^{(1)}, \hat{x}^{(2)}, \ldots, \hat{x}^{(n)}\right)$ is prediction value, and $N$ is the length of the sequence, then:

$$\text{MAE}(\boldsymbol{x}, \hat{\boldsymbol{x}}) = \frac{1}{N} \sum_{i=1}^{N} \left| x^{(i)} - \hat{x}^{(i)} \right|$$

$$\text{RMSE}(\boldsymbol{x}, \hat{\boldsymbol{x}}) = \sqrt{\frac{1}{N} \sum_{i=1}^{N} \left( x^{(i)} - \hat{x}^{(i)} \right)^2}$$

$$\text{MAPE}(\boldsymbol{x}, \hat{\boldsymbol{x}}) = \frac{1}{N} \sum_{i=1}^{N} \left| \frac{x^{(i)} - \hat{x}^{(i)}}{x^{(i)}} \right|$$

