# OpenReview forum: "Layerwise Recurrent Autoencoder for General Real-world Traffic Flow Forecasting"
_ICLR.cc/2019/Conference_

### Official Review · AnonReviewer3 · 2018-11-01
**Potentially interesting, though large omissions make this difficult to follow**

**Rating:** 3
**Confidence:** 4

**Review:**

The paper uses a number of deep learning approaches to analyse sets of Traffic data. However, as these sets of traffic data are never explained it is difficult to follow or understand what is going on here.

Some major comments:
1) Many of the key concepts in the paper are not discussed. The primary one would be that of what the two data sets contain. Without knowledge of this it is difficult to ascertain what is going on.

2) Many of the processes used are not described in enough detail to either understand what is going on or to re-produce the work. Without this it is difficult to make headway wit the work.

3) It is not clearly articulated what the experiments performed are doing. For example, how have you applied the other techniques to this data?

4) Key terms are not defined. Such as Traffic Flow.

5) The English structure of the paper is poor with many mistakes. A thorough proof-reading is essential.

Some more specific points:
- "with the larger road network, the difficulty of flow forecasting grows." - This seems to be a consequence of the other ones not a challenge in it's own right.

- What is "superiority"?

- "Spatiotemporal traffic flow forecasting task is currently under a heated discussion and has attracted a large research population." - evidence to back up this statement.

- Your contributions aren't contributions, but rather a list of what you have done.

- How does your related work relate to what you have done?

- Hard to parse "To extract temporal relationships within the history traffic flows, we model this process as a layering structure with autoencoder as cell"

- Appendices B and C should be in the main paper.

- What is in x^{(1)}?

- "When take the sparsity constrains into consideration" - what are the sparsity constraints?

- How do you obtain the weights?

- Figure 2 should come much sooner as it relates a lot of the concepts together.

- "On both datasets, we slice traffic flow information into 15 minutes windows, where 70% of data is for training, 10% for validation and remaining 20% for testing." - Is that each 15 mins is split 70:10:20?

- Proof by example is not a proof.

---

> ### Author Response · Authors · 2018-11-14
> **Reply for Reviewer3**
>
> Thanks for your comments. Here's our reply:
>
> For your major comments:
> 1)	The two datasets are introduced in the Appendix C, even with a distribution of sensors of the larger one. Please check it in our paper.
> 2)	We listed some key parameters in Appendix D, we believe with information of our paper, the readers can easily re-produce our work with Tensorflow. Of course, it would be easier if we make our code public, but the intellectual property right was kept by our company, and will be used in business projects. If the time is proper, the code would be released.
> 3)	The experiments are performed to show the better performance of our method over others, and the efficiency of our design of the network (SAE and GCN in Section 4.3), we think the experiments are clear in Section 4, including the other techniques (AR, ARIMA, ESN, LSTM and DCRNN). Please check our paper, especially in Section 4.
> 4)	Traffic flow is a common terminology in traffic-related forecasting area, which means the number of moving pedestrians or cars in traffic network.
> 5)	Yes, we agree that our language skills are poor, and needed to be improved.
>
> Specific points:
> 1)	It is natural in traffic forecasting that the more complex the road topology is the harder prediction would be. Since the complexity of traffic condition would exponentially grow with the larger traffic network.
> 2)	The superiority means "better performance", that we originally use driving distance to build spatial dependency, and our model is generalized for traffic forecasting in both large and small datasets.
> 3)	Yes, this sentence should be supported with citations.
> 4)	Sorry for giving you this impression, might because of our English skills, but this paper do have some novel contributions. This paper is focused on the design of the model and the originality of our model. We also compared the proposed model with other baselines, to show the efficiency and generalization.
> 5)	The related works are common forecasting methods and classified in two categories: statistic methods and deep learning methods. We selected some representative methods in both categories as baselines to compare with our model. And we also used some ideas from the related works (Diffusion convolutional recurrent neural network: Data-driven traffic forecasting and Deep Spatio-Temporal Residual Networks for Citywide Crowd Flows Prediction)
> 6)	To extract temporal relationships within the history traffic flows, we took SAE in the layer-wise structure as a basic unit to extract stable high-level temporal features, which is the preparation of traffic prediction.
> 7)	We put Appendix B in the end because we think the structures of autoencoder and SAE should be familiar in the forecasting field, and only for reference. The reason Appendix C was put behind the main body is because we want to keep the integrity and fluency in the experiment section.
> 8)	This is our fault, x^{(1)} is the input sequence of autoencoder, we should make it more clear.
> 9)	Sparsity constrain is sparsity, but in another presentation. When sparsity constraints are added to the objective function, an autoencoder becomes a sparse autoencoder, which considers the sparse representation of the hidden layer. Sparsity is used to avoid over-fitting.
> 10)	The weights can be obtained by some API of Tensorflow, and during the training process, they are trained by BP method.
> 11)	We think it would be better for reading experiment that reader can know the structure of model, before he sees it.
> 12)	Take the large dataset as example, we have 365 days of traffic data, and the first 80% are randomly divided into training sets and validation sets with a ratio of 7:1, i.e. 255 days for training, and 37 days for evaluation. The remaining 20% (73 days) is used as test sets. We believe this division method is closer to real use.
> 13)	We don't get your point in your comment, could you please make it clear?
>
> Hope our explanations can help you to understand our paper.

---

### Official Review · AnonReviewer2 · 2018-11-03
**significant clarification needed**

**Rating:** 5
**Confidence:** 3

**Review:**

This paper has potential, but I do not think it is ready for publication. I will ask some questions / make some suggestions:

1) Your first sentence makes a claim about there being a large body of research on traffic flow forecasting. I don't doubt this, but you should cite some papers, please.

2) Your contributions raise the following questions for me:

- Contribution 1 is that you use a very large dataset (for training? you don't say.) and a small dataset (for testing), thus proving that your method works and generalizes. Your method may be effective, but compared to what? Your method may generalize, but how do we know that if you've only tested it on one small dataset?

- Contribution 2 says that you creatively used lagged data in a time series model. This is probably a good idea, but it does not sound all that creative to me, compare with, e.g. an AR model.

- Contribution 3 says that you use driving distance to model spatial correlation. Again, this is probably a good idea, and when we get further we learn that you applied a Graph Convolution Network. Were these the choices that you claim are novel? Are they novel? What other choices might be reasonable and how would they compare?

3) Section 3 immediately jumps into the use of autoencoders. But I think you need to justify why we care about using autoencoders in the first place. If the problem is traffic forecasting, why don't you tackle that problem head on?

4) Section 3 mentions sparsity without justifying why I care about sparsity. This might be an important tool for regularization in a deep neural network. Or it might not be--given enough data and other regularization techniques (weight decay, early stopping, dropout).

5) Is the spatial dependency that you end up learning qualitatively different than the spatial dependency you would get by instead assuming a particular parametric form as is done in kernel methods / Gaussian processes, e.g. the Gaussian kernel or the Matern kernel parameterizes the covariance between observations at two spatial locations?

6) In your experiment I believe you randomly split 15 minute blocks into train/test/validate. I think this evaluation will be over-optimistic insofar as if 10:30-10:45 and 11:00-11:15 are in the train set, but 10:45-11:00 is in the test set, it will be relatively easy to predict 10:45-11:00. I would suggest considering train/test/validate splits based on larger chunks, e.g. leave the data in 15 minute blocks, but randomly select hours (4 blocks) to put in train/test/validate.

---

> ### Author Response · Authors · 2018-11-14
> **Reply for Reviewer2**
>
> Thanks for your review comments. Here's our reply:
>
> 1)	You are right, we should cite some publications to support it.
>
> 2)	As for our contribution one, the purpose we take experiments on both large and small datasets is to test the generalization of our network. Therefore, we trained our model on both large and small datasets, and tested with these two datasets. We compared our outputs with many baselines in Section 4, on both datasets, please check this in our paper.
>
> As for contribution two, we employed three different time series sequences to provide multi-perspective fusion learning. With the layer-wise structure, the daily and weekly periodic features of traffic volume are captured. In another word, a stack of historical sequence data is used in our experiments, rather than "lagged" time data.
>
> In the contribution three, we used driving distance to determine the spatial dependency, and this contribution was novel. And with the driving distance, we built a graph-based matrix, as a result, we then applied GCN to learn the inner features from the graph-based matrix. The use of GCN was inspired by one of our baselines, DCRNN (Diffusion convolutional recurrent neural network: Data-driven traffic forecasting). In DCRNN, they designed an experiment of comparing their diffusion method with the use of GCN. In their experiment, the result of diffusion method was better than GCN, but their dataset was not so large as ours, and when we compare their method with our datasets, the performance was not so good, even in our smaller dataset. Hence, we think the application of GCN in forecasting task is also novel.
>
> 3)	The reason why SAE is the first operator in the network is to extract stable high-level temporal features. We put this part in the beginning of Section 3. And the use of SAE do help the performance of forecasting, please check the content in Section 4.3.
>
> 4)	The sparsity of SAE was a proved solution of overfitting in practice. And the key of our paper was to provide a novel method for general forecasting task, therefore, we don't put much words on other choice, but just use sparsity as a common knowledge.
>
> 5)	The spatial dependency was determined by driving distance in our paper, and that should be a certain process, rather an assumption. Since the aim of our model was to provide accurate prediction in general situations, and the spatial dependency is crucial, since the spatial dependency is variable in different places. With driving distance and GCN, the model can grasp the most precise priori knowledge of spatial locations.
>
> 6)	In fact, we select the time based on a scheme, not randomly. Take the large dataset as example, we have 365 days of traffic data, and the first 80% are randomly divided into training sets and validation sets with a ratio of 7:1, i.e. 255 days for training, and 37 days for evaluation. The remaining 20% (73 days) is used as test sets. We believe this division method is closer to real use.
>
> If you have further questions, we are happy to discuss with you.

---

### Official Review · AnonReviewer1 · 2018-11-05
**Confused...**

**Rating:** 4
**Confidence:** 3

**Review:**

I am sorry but I am super confused with this paper. There is no clarity and about half of the sentences are written with broken english.

The model (as far as I can understand from the partial explanations and Figure 2) looks like a kitchen sink -- a combination of pieces from previously explored methods in the context of traffic flow estimation. This might be fine, but there is no motivation provided for this.

Rather than spending the method section with repeating well known Loss equations, KL-divergence, convolution, etc... Please focus on the architecture provided in the paper and the motivations behind it. More importantly, how it differs from previous approaches and why these choices have been made.

This paper is not ready for publication. It needs a re-write at least, preferably working out the original motivations behind architectural choices.

---

> ### Author Response · Authors · 2018-11-14
> **Reply for Reviewer1**
>
> Thanks for your review and sorry for the confusions.
>
> In the proposed model, there are three main parts and the motivations behind them are listed here:
>
> SAE: Regardless of the complex traffic trend, the SAE can stably extract implicit high-level features, which have almost the same characteristics as the original input. Besides, the general purpose of using SAE is to reduce the data dimension, this is also a motivation of exploiting SAE. Proved by experiments in Section 4.2, with SAE, the loss value of the model can drop quickly.
>
> RNN: As the commonly used method on time series prediction, RNN was used to process the implicit high-level features extracted by SAE. By the way, we used a layer-wise structured model, which was inspired by Deep Spatio-Temporal Residual Networks for Citywide Crowd Flows Prediction. The structure provided the model a conception of multiple perspectives, that considered three sequence of history data (current time volume, the volume of the time one week ago and the volume of the time four weeks ago). The model can not only learn the peak-valley periods of traffic in days, but learn the peak-valley periods in weeks.
>
> GCN: The motivation of using GCN was because the use of driving distance. Originally, we take driving distance, rather than Euclid distance to build spatial dependency. And based on driving distance, a graph-based matrix was built, which records the correlation between different sensors. As a result, graph convolutional network is used to learn prior knowledge in graph-based matrix, and it is regarded as an important factor in traffic forecasting. In Deep Spatio-Temporal Residual Networks for Citywide Crowd Flows Prediction, the authors used the traffic volume in region blocks to forecast traffic volume, but didn't consider the spatial correlations.
>
> These are our motivation of proposed model, if you have further questions, please let us know.

---

### Author Response · Authors · 2018-10-10
**Announcement of a typo**

We found a typo that may lead misunderstanding in the table 1, the MAE value of DCRNN and our LRA in ST-WB dataset with 15 min window should be 34.24 and 30.56 relatively.

Sorry for the ambiguousness caused by our fault, if you have any other questions, please feel free to ask us.

---

### Meta-Review · Area_Chair1 · 2018-12-14
**lack of clarity and precision in writing**

**Confidence:** 5
**Recommendation:** Reject

**Metareview:**

The paper proposes an interesting neural architecture for traffic flow forecasting, which is tested on a number of datasets. Unfortunately, the lack of clarity as well as  precision  in writing appears to be a big issue for this paper, which prevents it from being accepted for publication in its current form. However, the reviewers did provide valuable feedback regarding writing, explanation, presentation and structure,  that the paper would benefit from.